# The Socioeconomic Impact of Tourism in East Kazakhstan Region: Assessment Approach

**Selvina Aliyeva** [1,2,3,4] , **Xi Chen** [1,2,*], **Degang Yang** [1,2], **Kanat Samarkhanov** [1,2,3,4], **Ordenbek Mazbayev** [3], **Aday Sekenuly** [5], **Gulnura Issanova** [4,6] **and Sadyrbek Kozhokulov** [1,2]

[1] State Key Laboratory of Desert and Oasis Ecology, Xinjiang Institute of Ecology and Geography, Chinese Academy of Sciences, Urumqi 830011, China

[2] University of Chinese Academy of Sciences, Beijing 100049, China

[3] Faculty of Natural sciences, L.N. Gumilyov, Eurasian National University, Nur-Sultan 010008, Kazakhstan

[4] Faculty of Geography and Environmental Sciences, Al-Farabi Kazakh National University, Almaty 050040, Kazakhstan

[5] Faculty of Economics, L.N. Gumilyov, Eurasian National University, Nur-Sultan 010008, Kazakhstan

[6] Research Centre for Ecology and Environment of Central Asia (Almaty), Almaty 050060, Kazakhstan

[*] Correspondence: chenxi@ms.xjb.ac.cn; Tel.: +86-991-3835953

**Abstract:** The article presents a methodical approach based on an integrated assessment of the social and economic impacts of tourism for East Kazakhstan Region. The assessment was done using indicators such as the number of tourists, accommodation units, tourism facilities' billing, and other statistical data for the period of 2009–2018. Integrated assessment performed using weighted sums of considered parameters and ordinary least squares linear regression method is applied for effectiveness prediction. Applied approaches of arithmetic (calculated) and multivariate regression modeling of the integral tourism efficiency demonstrated the same results, which mean the approach can be transferred and applied for other regions of Kazakhstan. Growing trends in tourism efficiency are derived and conclusions made on their importance for regional development. The economic and social impacts of tourism in East Kazakhstan Region increased significantly during the period from 2003 to 2018.

**Keywords:** tourism; Kazakhstan; social efficiency; economic efficiency; integral assessment; regression

## 1. Introduction

Tourism is a globally growing industry that is widely recognized as an economic direction which is contributing significantly to global GDP [1,2]. Starting from the 1970s, tourism has become one of the most dynamically and intensively growing global industries [3–7]. In 2015, international mobility and tourism income were estimated to exceed 1 billion and 1.2 trillion US dollars, respectively, compared to 25 million and 2 billion US dollars in 1950 [8]. In recent years, new destinations are becoming interesting travel routes for international tourists, including several developing countries, which significantly contribute to the country's economy. In this regard, the popularizing of tourism and its role in sustainable development had great importance for these countries in recent decades [6,9–13].

Regional monitoring of domestic tourism is necessary for the timely detection of problems and to highlight the solutions that will accelerate its development. The evaluation of tourism's socioeconomic efficiency using a set of indicators is very important and leads to the development of possible strategies of inbound tourism as an economic direction. Indicators must meet certain requirements: an unbiased assessment based on statistical data, exclusion of duplicated data, data accessibility.

In most cases, evaluation of tourism's socioeconomic efficiency requires detailed data which are not fully accessible in such newly developing countries as Kazakhstan as they were not included in

the national statistics. Social and economic development indicators in this young country with only 28 years of history being published mostly at the regional level defines the restrictions for a detailed analysis of tourism impact. Therefore, we were forced to consider methods which could be adapted using existing official statistical data.

## 2. Literature Review

Due to the role of tourism in the economic development and economic expectations of different countries regarding tourism, there is substantial literature on this topic.

The main dimensions of tourism effects, as defined by the United Nations, include economic, socioecological, infrastructure sustainability, and attractiveness [14]. Along with this, methods for assessing economic and social effects of tourism were proposed in conditions of incomplete or incompletely reliable information. According to Song et al. [12], several methods were applied for estimating the effects of tourism on economics such as Keynesian-type multipliers [15], cost-benefit analysis (CBA) [16], Input-Output models [17,18], computable general equilibrium (CGE) frameworks [19–21], and tourism satellite accounts (TSA) [22]. On the other hand, the social impacts of tourism could be described by keywords like "host residents", "social impacts", "perceptions", and "attitudes", which refer to a qualitative assessment approach. In this regard, Deery et al. (2012) [23] consider the assessment of social effects from the temporal dynamics point of view, and outlined four main stages: definitions and concepts [24], modeling [25], assessment tool design [26], and assessment tool testing and adjustment [27]. Social surveys were one of the methods for the analysis of social indicators [28–30].

Similar methods were earlier published mainly by authors from the Commonwealth of Independent countries (CIS), which had the same situation of data scarcity or heterogeneity [31,32]. Thus, the comprehensive qualitative estimation approach was selected for the description of the socioeconomic effects of tourism based on the statistical data of East Kazakhstan Region and this kind of approach was not previously applied to assess the tourism impact in Kazakhstan.

East Kazakhstan Region (EKR) is one of the regions of the republic where the steadily growing quantity of internal tourists is almost 16 times more than the number of foreign tourists. It is among the top 5 regions according to inbound tourist number with an average of 16,565 persons and among the top 3 regions by domestic tourists visiting this area (~300 ths. persons). Thus, the EKR occupies one of the leading positions among the regions of Kazakhstan in terms of potential opportunities for the development of the tourism industry (Figure 1).

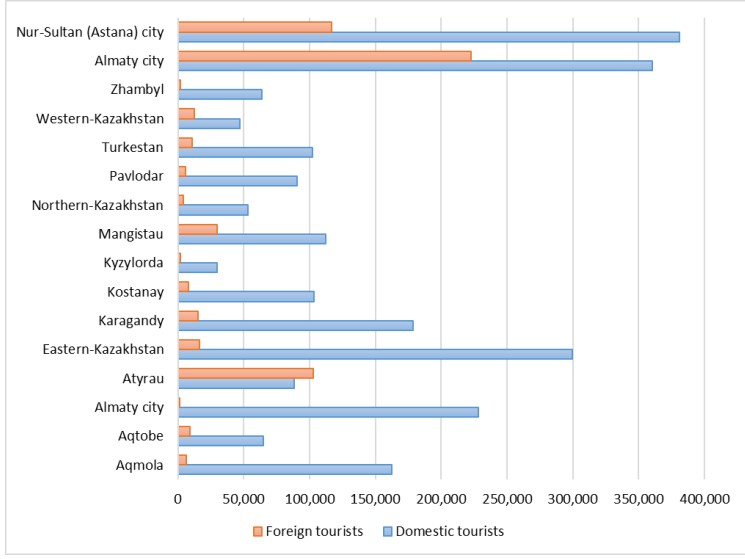

**Figure 1.** The average number of tourists hosted by accommodation facilities in Kazakhstan.



## 3. Materials and Methods

### 3.1. Study Area and Its Tourism and Recreation Resources

Kazakhstan is a land of the ancient Central Eurasian civilization with a marvelous and unique history. There was an intertwining of nomads and settling of the world's powerful empires and states, which emerged and perished for centuries. Important trade routes passing through this land-linked the East and the West. The country is rich in its variety of natural, cultural, and geographical attractions. Tourism plays a significant role in the global tourism and hospitality industry, and according to forecasts of experts, including the World Tourism Organization, its growth rates will continue to be high, and the income generated will make a significant contribution to the development of economies of various countries of the world [33]. However, tourism development remains a relatively new sector in Kazakhstan. Among the obstacles to the development of tourism in Kazakhstan are the following: inadequate price-quality ratio in comparison with foreign competitors; lack of demand for year-round tourist services; low level of competition in the tourism services market and in related industries; legislative regulation of the tourism industry and services that do not meet the requirements; difficult access to attractions; insufficient number of qualified specialists in the tourism sector [34].

Kazakhstan is the largest Central Asian country that tends to present its touristic opportunities globally. The contribution of tourism to GDP, according to World Travel and Tourism Council annual statistical report, was around 5.7% (about 9.3 million US dollars) in 2018. The number of tourists and income are increasing yearly, for example, 2 million foreign visitors spent 2.275 million US dollars in 2018 [35]. Currently, almost 490 thousand people or 5.7% of total employment are employed in the touristic sector. Table 1 shows the increase in inbound tourists hosted by accommodation facilities in Kazakhstan during recent years [36].

**Table 1.** The number of tourists hosted by accommodation facilities in Kazakhstan.

|  | 2013 | 2014 | 2015 | 2016 | 2017 | 2018 |
|---|---|---|---|---|---|---|
| Foreign tourists | 586,038 | 679,018 | 692,213 | 722,515 | 891,911 | 830,922 |
| Domestic tourists | 2,721,714 | 3,125,429 | 3,110,012 | 3,495,267 | 4,387,495 | 4,695,942 |

We can see that the number of foreign visitors exceeded 40% growth, while domestic tourists exceeded 70% growth in 2018 compared to 2013.

The same time, in the international market, Kazakhstan occupied the 85th position in 2013 according to the World Tourism Forum's Travel and Tourism Competitiveness Index [37]. This rating presented the results of a study of the competitiveness of 141 countries on 14 key parameters (Table 2).

**Table 2.** Competitiveness rating of Kazakhstan's tourism in the world market.

| Competitiveness Parameter | Rating |
|---|---|
| Business climate | 44 |
| Safety and security | 72 |
| Health and hygiene | 7 |
| HR and the labor market | 37 |
| The readiness of information and communication technologies | 48 |
| Priority of tourism | 84 |
| International openness | 124 |
| Price competitiveness | 49 |
| Environmental sustainability | 91 |
| Air transport infrastructure | 76 |
| Land and sea infrastructure | 102 |
| Tourist service infrastructure | 81 |
| Natural resources | 111 |
| Cultural resources and infrastructure for business tourism | 101 |

It can be observed in the table that the highest positions of the country were health and hygiene, business climate, the readiness of information and communication technologies, and price competitiveness. East Kazakhstan Region connects Southern Siberia, Altay, with Central Asia, and is bordered by two regions of Russia—one Chinese province and 3 Kazakh regions [38]. More than 40% of the surface water resources are concentrated here as there are approximately 885 rivers of more than 10 km length and about 1000 lakes with an area of more than 1 Ha in EKR [39]. The main waterway is the Irtysh with its tributaries (4248 km long, 1311 km of which flows within the region) and the large reservoirs with hydropower plants such as Bukhtyma, Shulba, and Oskemen that were built along with it [40]. The largest lakes are Alakol (2650 km$^2$), Zaisan (1800 km$^2$), Sasykkol (736 km$^2$), and Markakol (455 km$^2$). The climate is sharply continental with an average temperature in January of −16 to −20 °C, in July of 20–23 °C, and in the mountains of 16–18 °C. Mean annual rainfall varies from 150 to 1000–1500 mm. Natural protected zones such as Markakol and West Altay Nature Reserve, Katon-Karagay State National Natural Park, and the Semey Ormany State Forest Reserve are also located in EKR.

According to the national economic cluster development programs, the tourism cluster should become one of the progressively developing clusters in EKR. The concept of tourism industry development in the Republic of Kazakhstan before 2023 includes a plan for the development of the tourism cluster in EKR named "The Pearl of Altay" [41]. The city of Oskemen is defined as the center of the cluster, and key sites of tourist interest include eight objects (Figure 2):

(1)    The State Historical and Cultural Reserve Museum "Berel";
(2)    The Buqtyrma reservoir;
(3)    The Yertis River–Zaysan Lake;
(4)    The Katon-Karagay State National Natural Park;
(5)    The Kiyn-Kerish canyon;
(6)    The city of Ridder–The West Altay State Natural Reserve;
(7)    The Lake Alakol;
(8)    The city of Semey.

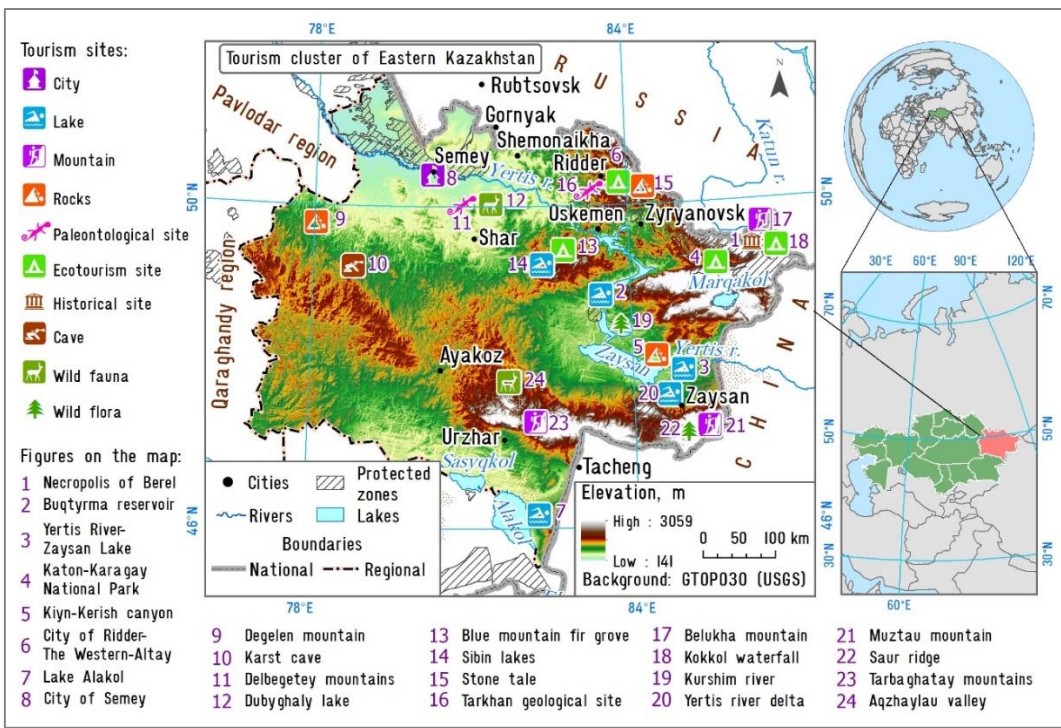

**Figure 2.** Map of the tourism cluster in East Kazakhstan Region (EKR).

According to the concept of tourism industry development in the Republic of Kazakhstan before 2023, the tourism cluster in East Kazakhstan Region is expected to become the center of ecological tourism, which could offer active and adventure tourism, recreation at mountains and lakes, medical and health tourism, spa treatments, etc. Besides the listed objects, EKR has tourist objects starting with number 9 on the map above [42]. Though it is not yet possible to place all tourist objects on the map, we tried to include the main sites, like mountains, lakes, geological objects, paleontological sites, ecotourist objects, wild fauna, wild flora sites, and national protected zones. Objects were placed on an elevation map received from the US Geological Survey's Center for Earth Resources Observation and Science (EROS) [43]. There are other tourism resources in East Kazakhstan Region, which could form the basis of regional development along with other branches of the economy.

## 3.2. Research Methods

The first part of the study consisted of online surveys devoted to the attractiveness of the whole Kazakhstan and East Kazakhstan Region for international and domestic tourists, or people with business purposes in order to depict the potential of tourism businesses in this country. Though the motivation, needs and preferences, and spending patterns of business tourists are different from leisure tourists, it was not possible to distinguish them. Queries were published through the web-surveying tools www.forms.google.com and www.wenjuan.com. They included 30 questions on nationality, employment, age, preferences, and informational awareness of Kazakhstan. They were provided in English and Russian languages, and the translation was verified by native language representatives. The online survey was advertised through both personal contacts and social networks. No reward was assigned for the respondents that participated in surveys and snowball sampling was the selected strategy. The surveys were conducted during October–November 2018 among citizens of different countries, its results are presented in the Results part of this paper. The purpose was to have a sense of how attractive Kazakhstan can be for tourism.

After that the tourism indicators recommended by UNWTO were selected for this study. They considered the supply of tourism, including accommodation for visitors, food and beverage serving activities, passenger transportation, travel agencies, and other activities, and the employment in the tourism industries [44,45].

In order to assess the socioeconomic impacts of tourism in those mapped areas, it was necessary to use statistical data published by the East Kazakhstan Statistics Department of Agency of Statistics of the Republic of Kazakhstan [46]. Indicators of tourist companies and hotel statistics were considered in this study, while the methodology for assessing socioeconomic efficiency was based on the following principles:

(1) The complexity of the assessment of indicators;
(2) Availability of the information used;
(3) Simplicity and replicability of the methodology, its correct reproduction in other areas of Kazakhstan;
(4) The possibility of obtaining a quantitative integral estimate, allowing comparative analysis for parts of the region.

Statistical data on economic and social indicators related to tourism for the period from 2003 to 2018 were collected (Figure 3) [47].

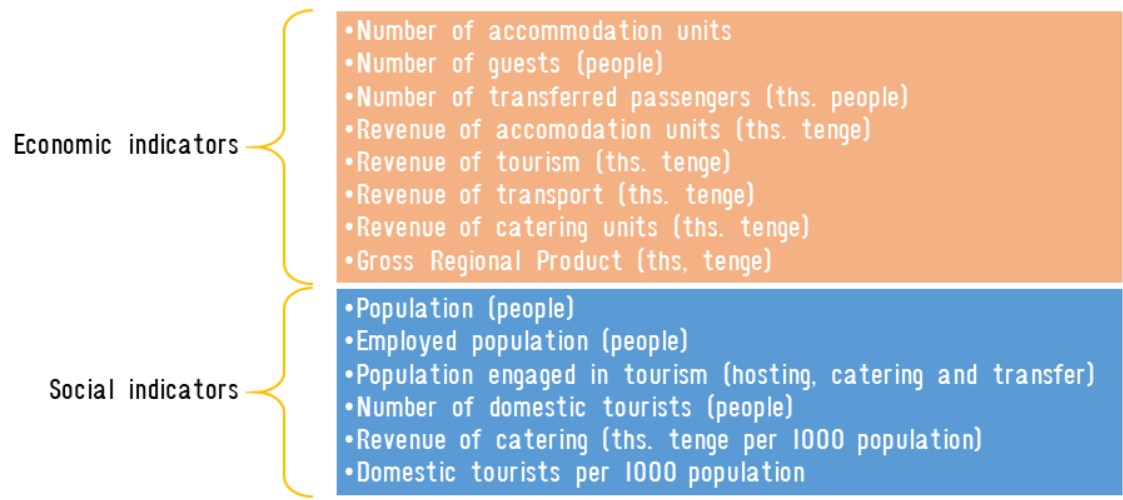

**Figure 3.** Indicators selected for the study.

Economic indicators included data on accommodation units, number of transferred passengers and revenues of tourism, transport, catering units, and gross regional product. Among other indicators included in the group of social indicators, the number of domestic tourists and revenue of catering per 1000 population were considered as affecting the social situation. The population engaged in tourism included all staff of accommodation units, tourism agencies, catering units, while the main part of this was the population involved in transporting system as far as it was normally considered as an infrastructural dimension of tourism.

The data processing during this study consisted of (1) calculating coefficients of selected statistical data, (2) calculating integral social and economic efficiency of the tourism, (3) modeling of integral tourism efficiency indicators, (4) comparing calculated and modeled efficiency indicators.

Thus, at the step of statistical data conversion into coefficients, all indicators were standardized [47] in order to estimate integral efficiency for each indicator using the following Equation (1):

$$k_{ti} = \frac{(k_i - k_{min})}{(k_{max} - k_{min})},$$ (1)

where $t = 1$ or $t = 2$, $k_{ti}$ is the $i$-th standardized indicator from block 1 (economic efficiency indicators) or 2 (social efficiency indicators), and it is equal to the ratio of the difference between its observed value $k_i$ for each time period and its minimum value $k_{min}$ to the difference between observed the maximum value $k_{max}$ and minimum $k_{min}$.

For calculation of the integral social and economic efficiency of tourism, the correlation between economic and social indicators was estimated according to the Equation (2).

$$r_{ij} = \frac{\sum_{ij}^{n}(x_n - \overline{x}) \cdot (y_n - \overline{y})}{\sqrt{\sum_{ij}^{n}(x_n - \overline{x})^2 \cdot (y_n - \overline{y})^2}},$$ (2)

where $r_{ij}$ stands for correlation coefficient between observed social indicators $x_n$ and economic indicators $y_n$ for each time period from $i$ to $j$, while $\overline{x}$ and $\overline{y}$ are the mean values of indicators.

After that, weights $w$ for each indicator were estimated as the ratio between the sum $\sum$ of correlation coefficients $r_{ij}$ of the selected indicator for time periods from $i$ to $j$ to the total of all correlation coefficients for a group of social or economic indicators respectively (3).

$$w = \sum_{i=1}^{n} r_{ij} / \sum_{i=1}^{n}\sum_{j=1}^{n} r_{ij}$$ (3)

Derived variables were used for assessment of the integral efficiency considered as the sum of products for each group according to Equation (4):

$$y_t = \sum_{i=1}^{n} w_i \cdot k_{ti},\qquad(4)$$

where $t = 1$ or $t = 2$, while integral efficiency $y_1$ is a group of economic efficiency indicators and $y_2$ is a group of social efficiency indicators of tourism, $k_{ti}$ is the $i$-th standardized indicator from block 1 or 2, $w_i$ is the weight of the $i$-th normalized indicator from block $t$, and $n$ is the number of indicators for each block of indicators.

Then, the integral social and economic efficiency of tourism $z$ can be estimated considering them as similar factors (5):

$$z = y_1 + y_2.\qquad(5)$$

Modeling of integral tourism efficiency was conducted in order to compare it with the previously calculated integral efficiency of tourism. For this purpose, the simple multiregression analysis was performed using initial statistical indicators (5).

$$Y = b_0 + b_1 x_1 + \ldots + b_n x_n,\qquad(6)$$

where $Y$ is integral tourism efficiency, $b$ is estimated values, and $x$ is tourism efficiency indicators from 1 to $n$. Model performance was described, the modeled and integral tourism efficiency were compared, and their mutual correlation was checked, and results were plotted.

## 4. Results

### 4.1. The Online Survey Results

The short online survey results involved people from different countries including citizens of Kazakhstan. A total of 433 answers were received, and among these, there were participants from China, Kazakhstan, Russia, West Asia, Europe, South East Asia, South Asia, Central Asia, and Africa (Figure 4) that answered the survey questions.

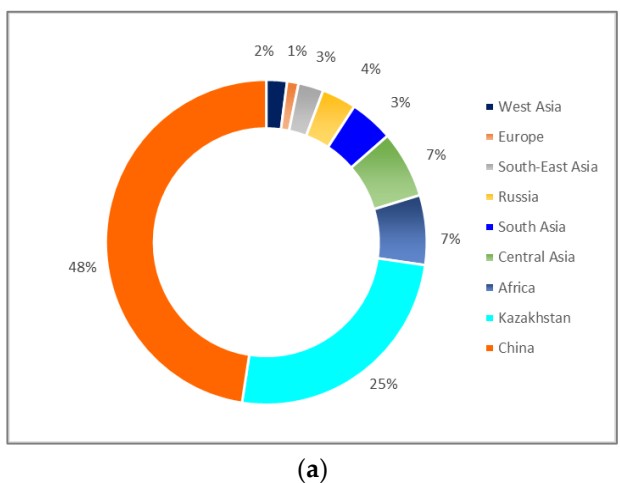
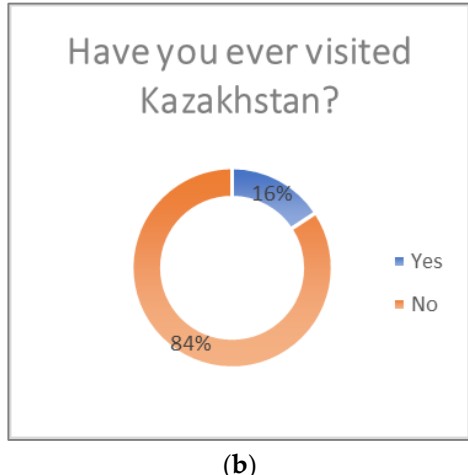

| (a) | (b) |

**Figure 4.** Main results of the online survey: (**a**) Geographic distribution of respondents; (**b**) Number of respondents who visited Kazakhstan.

Chinese respondents were represented by 206 persons, Africa by 30 respondents, Central Asia by 29, and other parts by 59 participants, while Kazakhstan was represented by 109 respondents. Among the 324 foreigners, 273 had preliminary information about Kazakhstan and only 16% (43 persons)

had visited this country, with most of the visits being business trips, and only 7 foreigners had visited East Kazakhstan. Moreover, 61% (67 persons) of Kazakh citizens had visited East Kazakhstan.

### 4.2. Tourism Efficiency Indicators in East Kazakhstan

Considered statistical data were converted into coefficients, all indicators were standardized according to Table 3.

**Table 3.** Efficiency indicators of tourism for each 5-year period from 2003 to 2018.

| Indicator | 2003 | 2008 | 2013 | 2018 |
|---|---|---|---|---|
| *Economic efficiency indicators* | | | | |
| Number of accommodation units | 0 | 0.34 | 0.48 | 1 |
| Number of guests | 0 | 0.31 | 0.62 | 1 |
| Number of transferred passengers | 0 | 0.28 | 0.87 | 1 |
| Revenue of accommodation units | 0 | 0.19 | 0.47 | 1 |
| Revenue of tourism | 0 | 0.1 | 0.53 | 1 |
| Revenue of transport | 0 | 0.09 | 0.41 | 1 |
| Revenue of catering units | 0 | 0.21 | 0.86 | 1 |
| Gross Regional Product | 0 | 0.19 | 0.59 | 1 |
| *Social efficiency indicators* | | | | |
| Population | 1 | 0.24 | 0.21 | 0 |
| Employed population | 0.22 | 0.78 | 0.62 | 0 |
| Population engaged in tourism | 0 | 0.23 | 0.49 | 0.78 |
| Number of domestic tourists | 0 | 0.1 | 0.64 | 1 |
| Revenue of catering per 1000 population | 0 | 0.21 | 0.85 | 1 |
| Domestic tourists per 1000 population | 0 | 0.1 | 0.63 | 1 |

The maximum value was 1 while the minimum value was equal to 0, which means the minimum value of the indicator was recorded for this period. This the table shows that values of considered indicators mostly had increased in 2018 while the population and the proportion of employment had decreased starting from 2015 and was like the initial level of 2003.

According to the workflow described in the previous section, correlation matrixes of economic and social efficiency indicators were estimated (Figure 5).

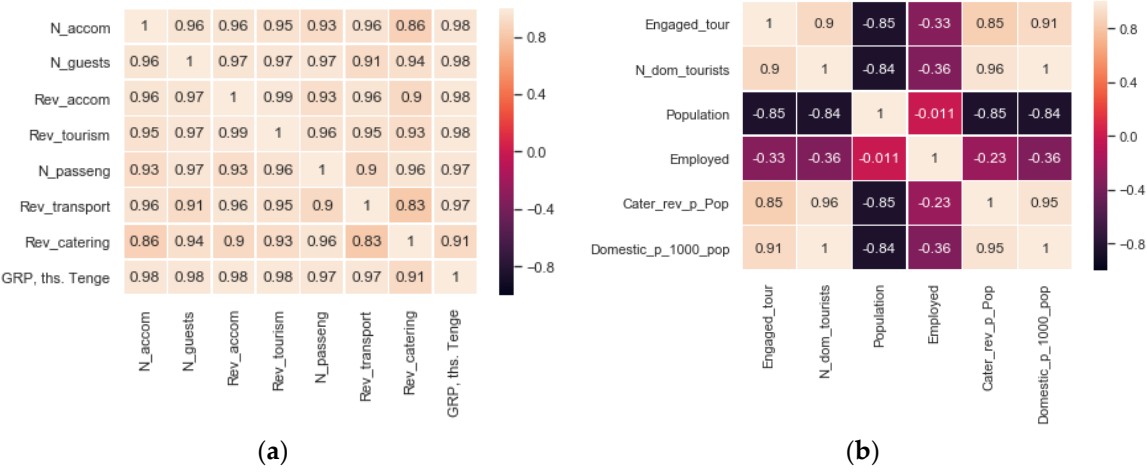

**Figure 5.** Correlation matrixes of (**a**) economic efficiency indicators and (**b**) social efficiency indicators.

Selected economic indicators demonstrated a strong positive correlation with the minimum value of 0.83 (revenue of catering units and revenue of transport). As for social indicators, four indicators had a strong positive correlation (minimum 0.85) while the population demonstrated a strong negative

correlation. The Employed population had low negative correlation coefficients with other indicators, for example, with a number of domestic tourists (Figure 6).

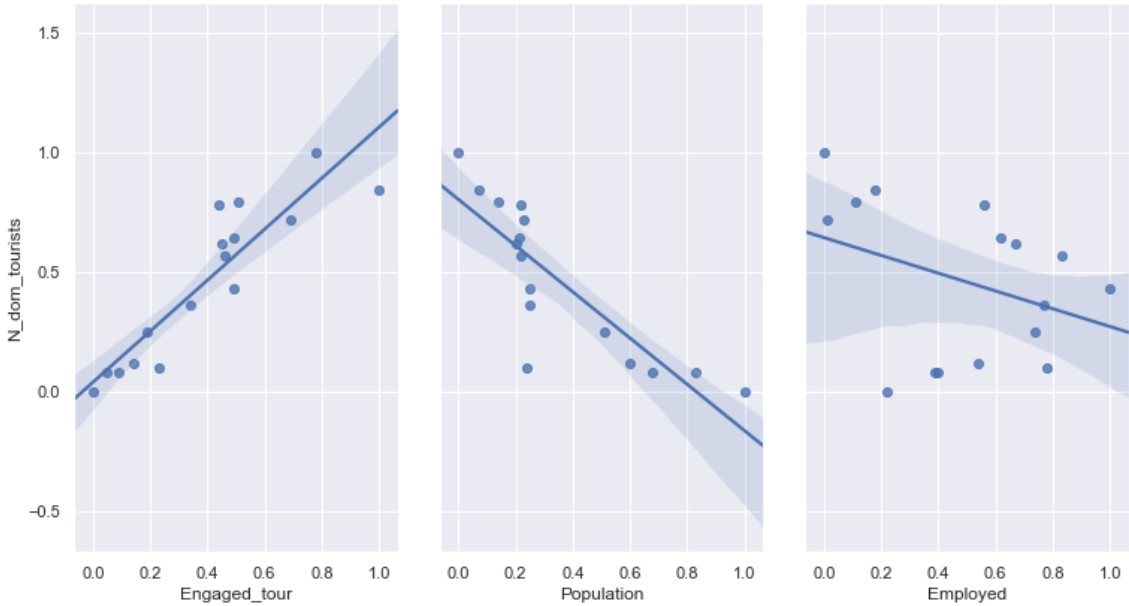

**Figure 6.** Fitted social efficiency indicators.

Weights of each indicator were calculated from the sum of derived correlation coefficients for each group of tourism efficiency indicators (Table 4).

**Table 4.** Weights of each indicator for each group of tourism efficiency indicators.

| Indicator | Weight | | Weight |
|---|---|---|---|
| *Economic efficiency indicators* | | *Social efficiency indicators* | |
| Number of accommodation units | 0.12 | Population | −0.31 |
| Number of guests | 0.13 | Employed population | −0.04 |
| Number of transferred passengers | 0.13 | The population engaged in tourism | 0.32 |
| Revenue of accommodation units | 0.13 | Number of domestic tourists | 0.34 |
| Revenue of tourism | 0.13 | Revenue of catering per 1000 population | 0.34 |
| Revenue for transport | 0.12 | Domestic tourists per 1000 population | 0.34 |
| Revenue of catering units | 0.12 | | |
| Gross regional product | 0.13 | | |

Weights for economic efficiency indicators had positive values while population and the employed population had negative values, and other social efficiency indicators demonstrated positive values. They were used for the calculation of weighted economic and social integral efficiency (Table 5). The linear increase in economic and social efficiency, as well as calculated integral efficiency, was observed during the period from 2003 to 2018.

All social and economic indicators were used as factors for modeling. A linear regression model was set up and input factors were fitted. Intercept and coefficients for each indicator of the linear regression model were calculated. Arithmetic (calculated) and modeled integral accuracy coefficients were compared, results for every period matched with automatically predicted values of the model, and calculated and modeled integral accuracy coefficients had the same values, and their mutual correlation was maximum. In the case of integral efficiency, it started with a negative value of 0.32 and significantly increased above 0 in 2007 compared to the previous years, but in 2008, had the same value. Between 2010 and 2014, a straight increase could be observed followed by small changes in

2015. The integral efficiency of tourism impact in East Kazakhstan Region had an increasing trend and calculated and modeled integral accuracy coefficients had almost the same pattern as presented in Figure 7.

**Table 5.** Dynamics of economic and social efficiency.

| Year | Economic Efficiency | Social Efficiency | Calculated Integral Efficiency | Modeled Integral Efficiency |
|------|---------------------|-------------------|-------------------------------|------------------------------|
| 2003 | 0 | −0.32 | −0.32 | −0.32 |
| 2004 | 0.04 | −0.2 | −0.16 | −0.16 |
| 2005 | 0.06 | −0.12 | −0.06 | −0.06 |
| 2006 | 0.12 | −0.05 | 0.07 | 0.07 |
| 2007 | 0.2 | 0.11 | 0.31 | 0.31 |
| 2008 | 0.21 | 0.1 | 0.31 | 0.31 |
| 2009 | 0.24 | 0.34 | 0.58 | 0.58 |
| 2010 | 0.37 | 0.57 | 0.94 | 0.94 |
| 2011 | 0.46 | 0.63 | 1.09 | 1.09 |
| 2012 | 0.53 | 0.71 | 1.24 | 1.24 |
| 2013 | 0.61 | 0.79 | 1.4 | 1.4 |
| 2014 | 0.71 | 0.87 | 1.58 | 1.58 |
| 2015 | 0.71 | 0.92 | 1.63 | 1.63 |
| 2016 | 0.81 | 0.94 | 1.75 | 1.75 |
| 2017 | 0.84 | 1.1 | 1.94 | 1.94 |
| 2018 | 1.01 | 1.27 | 2.28 | 2.28 |

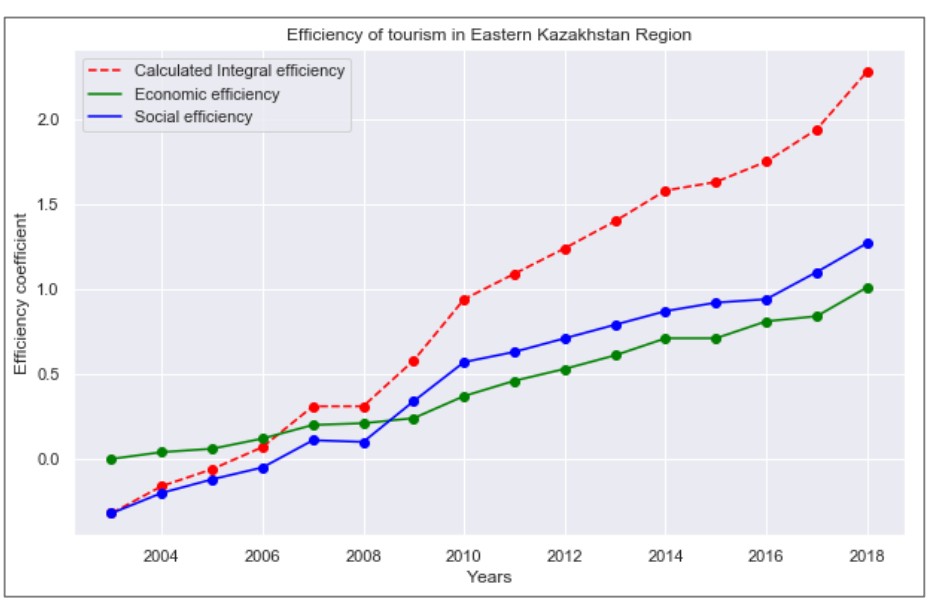

**Figure 7.** Integral efficiency of tourism impact in East Kazakhstan Region.

As can be observed in the figure, economic efficiency coefficients had a slightly growing pattern, while social efficiency had low increases in 2008 and 2015. We can see that the dynamics of integral tourism efficiency shows a significant increase in 2010. They were all reflected in the pattern of the integral efficiency of tourism, while the main trendline of both social and economic efficiency had a straight growing plot.

## 5. Discussion

The findings of this study show that the effects of tourism in East Kazakhstan Region could be interpreted as positively impacted factors in the economic and social dimensions of regional development. However, we need to remember that the analysis done during this study does not

necessarily imply a causal relationship between considered indicators. Our own short survey results show that though 84% of foreigners had preliminary information about Kazakhstan, only 16% of them had visited this country. However, most of these were business trips, and only 7 foreigners have visited EKR. Moreover, only 61% of Kazakh respondents had visited East Kazakhstan. This means that foreigners do not know Kazakhstan or EKR as a possible tourism destination.

The social and economic efficiency of tourism in EKR demonstrate strong growth with a period of stagnation in 2007–2008 followed by a significant increase in 2009. The similar values of the integral coefficients are possibly evidence of the Global financial crisis with a sharp crash in 2007 caused by "excessive liquidity creation by the world's two leading central banks" among other reasons [48]. Other comparatively slight increases in 2015 could be due to the Kazakh national currency default regarding prices for exported products [49], which impacted both the population and employment. Among other indicators, there were two which had negative coefficient values, namely population and employed population. They directly reflect macroeconomic changes as the population had decreased in 2018 becoming equal to or having almost the same value as in 2003. Therefore, these indicators had negative correlation coefficients with other social efficiency indicators. Moreover, the employed population had correlation coefficients lower than 0.5. This can be due to low numbers of staff in tourism area compared to a commonly employed population of the study area.

On the one hand, this study provides an opportunity for depicting tourism effects at regional or national levels for a broad number of researchers, due to the accessibility of statistical data, especially in the such a newly independent state as Kazakhstan.

On the other hand, these indicators do not necessarily reflect the real tourism situation in the study area as this study was limited by data quality. The considered data are presented only at the regional level and only for 1-year periods, which means it is impossible to assess and monitor of tourism at a more detailed level, for example in districts or settlements, tourism site levels. Changes in the seasonal trends of tourism are not assessable as well. The existing recording and classification of Kazakh national statistics should be changed in order to monitor tourism properly by adding back several categories, such as tourism agency statistics, which are excluded from national records.

East Kazakhstan Region has all the opportunities to become more famous for its ecological, historical, and health recreational resources, which can be its "business card" for neighboring countries such as China, Russia, and Mongolia, with respect to distance. Also, several measures can be recommended: organizing international flights for neighboring country citizens, international tourism routes, facilitation of a visa regime for foreign tourists, organizing low-cost flights for domestic tourists, tax breaks for tourism activities in order to decrease charges, marketing of tourism objects, etc.

Applied approaches on the calculated and multivariate regression modeling of integral tourism efficiency demonstrated the same results, which means the approach can be transferred and applied for other regions of Kazakhstan and any developing country. This is useful for tourism monitoring in the case of a lack of input data, such as in Kazakhstan. The applied methods of the social and economic impact of tourism give a more quantitative rather than qualitative dimension, which could be something to be examined in future research works.

## 6. Conclusions

This study shows that the economic and social impacts of tourism in East Kazakhstan Region increased significantly during the period from 2003 to 2018. Selected social and economic indicators were checked for mutual correlation and demonstrated high values, except the employed population. The employed population had correlation coefficients lower than 0.5, which can be related to the fact that other economic branches occupy most positions in common employment of the region comparing tourism. Also, the population had negative correlation values which can be explained by immigration and emigration processes in the study area caused by different reasons including economic situation.

Though the pattern of integral efficiency of tourism on economic and social indicators is negative at the initial observation period it turned out to become positive in the following years. Macroeconomic

events, such as the financial crisis of 2007–2008 and the default of the Kazakh currency in 2014–2015 are reflected in social and economic indicators and, as a result, in the integral efficiency of tourism.

Applied approaches on arithmetic (calculated) and multivariate regression modeling of the integral tourism efficiency demonstrated the same results, which means the approach can be transferred and applied to other regions of Kazakhstan and any developing country.

East Kazakhstan Region has all the opportunities for developing tourism and becoming one of the well-developed tourism centers attracting tourists from all over the world. One of the main reasons for EKR's underdeveloped tourism is the lack of correct internal policy including economic motivation measures for tourism entities. On the other hand, tourism in EKR needs improvements to common marketing strategy, product development, pricing regulations, etc. In order to improve the monitoring of tourism, it is recommended to adjust the recording and classification of existing Kazakh national statistics in order to monitor tourism records.

**Author Contributions:** Conceptualization, S.A., X.C., D.Y., O.M., A.S., K.S., G.I. and S.K.; Formal analysis, S.A. and K.S.; Methodology, S.A., X.C., O.M., A.S., K.S., G.I. and S.K.; Software, K.S.; Supervision, X.C. and D.Y.; Visualization, K.S.; Writing—original draft, S.A., K.S., G.I. and S.K.; Writing—review & editing, X.C., D.Y., O.M. and A.S.

**Funding:** This research was funded and supported by: 1. the National Science Foundation of China, Pan-Third Pole Environment Study for a Green Silk Road (NSFC 41761144079; Pan-TPE XDA20060303). 2. The Xinjiang Institute of Ecology and Geography of the Chinese Academy of Sciences.

**Acknowledgments:** The authors would like to thank Dr. Tokhir Bozorov (Institute of Genetics an Plant experimental Biology, Tashkent, Uzbekistan) for useful ideas, reasonable suggestions.

**Conflicts of Interest:** The authors declare no conflict of interest.

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
