# Peer review of "The Socioeconomic Impact of Tourism in East Kazakhstan Region: Assessment Approach"

_sustainability, doi:10.3390/su11174805_

Round 1
Reviewer 1 Report
I am reading this manuscript from the perspective of a tourism expert. Even though the economic analyses and statistical accuracy is unquestionable, I have many questions as to how authors can claim tourism development in the Eastern Kazakhstan Region is the most suitable development strategy.
First and foremost, this manuscript lacks purpose and there is incongruence between the findings and recommendations. It is also difficult to connect the analyses of the research in the context of tourism development in the Eastern Kazakhstan Region. The Introduction started with the UNSDG, however, neither the analyses or the recommendations have anything to do with the UNSDG or sustainability issues in Kazakhstan. For these major flows, I recommend rejecting this paper for the Sustainability journal and suggest that the authors submit the manuscript to a journal in discipline of economics.
Below are the comments the authors can use to revise and improve the manuscript.
It is unclear from the abstract and the introduction sections what this research is really aiming at. If it is just to prove a certain methodology of economic analyses, the abstract and introduction sections must emphasise this point. As it reads now, it sounds like the purpose of the research is an economic assessment for the potential of tourism development in the region, which is not attained in this manuscript.
The opening sentence of the Introduction section is grammatically correct, but it sounds awkward. In addition to the grammatical point, why did the authors select only three UNSDG? What is the significance of these 3 goals and why are not all of the 17 goals applicable to Kazakhstan? This must be explained to the readers.
In the Abstract, it is stated, “the approach can be transferred and applied for other regions of Kazakhstan and any developing country,” yet the manuscript fails to prove this point. Perhaps the results ‘can be’ applied to other regions in Kazakhstan where culture, politics and resources are shared with the Eastern Kazakhstan region. However, the authors neglected to explain the national characteristics of Kazakhstan, which can be shared with other ‘developing country’ (countries). It is known that some developing countries have the social capital of hospitality, which is suitable for tourism and hospitality development, while other countries do not appreciate the presence of outsiders and thus do not want to develop tourism and hospitality. Some of these issues are already apparent in Table 2 (p. 2) – i.e., International openness, Cultural resources and infrastructure for business tourism, Natural resources, Land and sea infrastructure are very low on world competitiveness as well as the Priority of tourism and Environmental sustainability. 

Where is the Literature Review Section (or Background Information for the Research) in this manuscript? There is no clear thesis statement, other than what is in the Abstract.
The readers must be convinced with the information from Table 2, which indicates that Kazakhstan is clearly not yet ready for tourism development that the authors now suggest “adventure tourism, recreation at mountains and lakes, medical and health tourism, spa treatment” (p. 3), as the “popularizing of tourism and its role in sustainable development have great importance” (p.1) for Kazakhstan. Having natural and geographical resources does not mean the area is suitable for tourism development.
This reviewer was puzzled by the authors’ suggestion to create a “centre for ecological tourism” using “tourism clusters”. What do the authors mean by this? How could “adventure tourism, recreation at mountains and lakes, medical and health tourism, spa treatment” or outdoor activities fit into a “centre for ecological tourism” while Kazakhstan’s competitiveness on “Environmental sustainability” is so low? It sounds like a perfect recipe for environmental devastation by developing unsuitable tourism products. The authors must provide thorough reasoning for this suggestion.
The evaluation of socio-economic efficiency in this research seems to be more focused on the economic side rather than the social side. In the research, the social dimensions such as infrastructure and governance development, and human capital development are never mentioned. From the analyses and findings, three social aspects (Population
Employed, Population, Population engaged in tourism) are not only declining over the years, but also have negative “economic x social” correlations (Correlation Matrix, p.7). Why did authors underplay such important factors and overplay economic efficiency? And how could the findings support UNSDG 8, 12 and 14 (“decent jobs”, “responsible consumption” and “life underwater” respectively)?
The research method section requires further explanation. There are two sets of data collection (1. online survey with potential tourists and 2. Statistical data from the East Kazakhstan Statistics Department of Agency of Statistics of the Republic of Kazakhstan). The first data collection method requires more information. For instance, (a) in what language(s) was the online survey conducted? If in multiple languages, how was the translation verified? (b) How was the online survey advertised? (c) How were the respondents recruited? As there any reward for participation in this survey? (d) What were the selection criteria of countries to conduct survey? (e) What was the sampling method? Stratified sampling? Snowballing sampling?
There is one statement “But as it was mentioned above, some data categories were omitted due to inaccessibility” (p.4). What does this mean and where was it mentioned above (about inaccessibility), as this is not covered above in the article? Only a brief mention is given that the UNWTO’s recommended tourism indicator (what are they comprised of?) was used, and there seems to be no justifications or explanations. 

The collected data was described by the nationalities of the respondents (p. 6). Why were nearly half of the total respondents (48%) from China and Kazakhstan respondents (potential domestic tourists) accounted for 25%? What are the reasons of this imbalanced respondents ratio? Was it purposeful? It is also stated, “most of the visits were business trips” (p. 7) but how could you apply the responses from business tourists to potential leisure tourists? It is known that the motivation, needs and preferences, spending patterns of business tourists are different from leisure tourists. What is the purpose of the online survey other than demographic data of respondents?
In the Conclusion section, background tourism information in Kazakhstan was summarised in Figure 7 (p.10). This information must be presented in the Literature review section after the Introduction (which is unfortunately missing from this manuscript).
Author Response
Thank you very much for your reasonable and useful comments.
We hope our answers match your questions.
Wish you the best,
Authors

Reviewer 2 Report
The title of the article is concise, clear and agrees with the subject and the contents.
The summary reflects the structure and contents of the article with precision and clarity.
The methodology has been established clearly and precisely and is adequate for the study carried out.
Some issues:
Introduction: information about the case studies is ok but the theoretical framework must be extended. Moreover, I miss a critical discussion of the indicator.
Conclusions: The link between the theory and analysis needs certainly to be strengthened.
Author Response
Thank you very much for your comments. We tried to proved answers and hope you can find them reasonable.
